# Neuropilin-1 as a Key Molecule for Renal Recovery in Lupus Nephritis: Insights from an NZB/W F1 Mouse Model

**DOI:** 10.3390/ijms252111364

**Published:** 2024-10-22

**Authors:** Sebastian Sandoval, Cristina Solé, Blanca Joseph-Mullol, Maria Royo, Teresa Moliné, Alejandra Gabaldón, Josefina Cortés-Hernández

**Affiliations:** 1Rheumatology Research Group—Lupus Unit, Vall d’Hebrón University Hospital, Vall d’Hebrón Research Institute (VHIR), Universitat Autònoma de Barcelona (UAB), 08193 Barcelona, Spain; sebastian.sandoval@vhir.org (S.S.); blanca.joseph@vhir.org (B.J.-M.); maria.royo@vhir.org (M.R.); fina.cortes@vhir.org (J.C.-H.); 2Department of Pathology, Vall d’Hebrón University Hospital, 08035 Barcelona, Spain; teresa.moline@vhir.org (T.M.); alegabaldon@hotmail.com (A.G.)

**Keywords:** systemic lupus erythematosus (SLE), lupus nephritis (LN), neuropilin-1 (NRP-1), biomarker, renal recovery

## Abstract

Systemic lupus erythematosus (SLE) is an autoimmune disease affecting multiple organs, with lupus nephritis (LN) occurring in 40–50% of SLE patients. Despite advances in diagnosis and treatment, LN remains a major cause of morbidity and mortality, with 10–20% of patients progressing to end-stage renal disease (ESRD). While knowledge of LN’s pathogenesis has improved, mechanisms of renal recovery are still largely unexplored. Neuropilin-1 (NRP-1), a transmembrane receptor expressed in renal tissue, has emerged as a promising biomarker for assessing renal recovery in LN. This study evaluates and correlates longitudinal levels of NRP-1 with kidney histology using an NZB/W F1 mouse model of LN. A total of 30 mice were used, with 15 receiving intravenous cyclophosphamide (CYC) and 15 being untreated. NRP-1 levels were measured in urine and serum, and kidney samples were taken from a subgroup of mice for histological evaluation. The results demonstrated a progressive increase in renal and urinary NRP-1 expression, particularly notable at weeks 26 and 32. Urinary NRP-1 levels above 34.40 ng/mL were strong predictors of favorable renal response, showing 100% sensitivity and 88% specificity. These findings indicate a robust correlation between urinary NRP-1 levels and renal histological recovery, underscoring the potential of NRP-1 as a valuable biomarker for assessing renal response in LN. This study demonstrates that renal production of NRP-1 is linked to histological recovery and establishes a foundation for future research into the role of NRP-1 in lupus kidney recovery.

## 1. Introduction

Systemic lupus erythematosus (SLE) is an autoimmune disease characterized by widespread inflammation and immune-complex-mediated damage affecting multiple organs, including mucocutaneous, musculoskeletal, and renal tissues [1,2]. Up to 50% of SLE patients develop lupus nephritis (LN) at some point during the disease course, with incidence rates varying depending on race, ethnicity, and access to medical care. The diagnosis of LN is typically based on clinical and laboratory parameters [3], though renal biopsy remains the gold standard for confirming diagnosis, assessing disease activity and chronicity, guiding treatment decisions, and prognosis [4].

Managing LN presents significant challenges in SLE, particularly due to the difficulty of diagnosing subclinical onset and identifying relapses before the development of severe complications. Only a small proportion of LN patients achieve complete renal response within the first 6–12 months of therapy. Renal flares are common during maintenance treatment, and 10–20% of patients progress to end-stage renal disease (ESRD) within five years of diagnosis [4]. To date, conventional clinical markers such as proteinuria, glomerular filtration rate (GFR), urinary sediment, autoantibodies to double-stranded DNA (anti-dsDNA), and complement levels lack the sensitivity and specificity needed to accurately detect kidney disease activity, predict early nephritis relapse, monitor progression to chronic kidney disease, or assess treatment response. As a result, there has been growing interest in identifying novel biomarkers for disease progression in recent decades [5].

Neuropilin (NRP-1) was initially identified as a co-receptor for class 3 semaphorins (SEMA3A), a family of molecules involved in axonal repulsion, cell apoptosis, cell migration, tumor suppression and progression, angiogenesis, and immune dysregulation [6]. Decreased NRP-1 and SEMA3A expression in serum B cells from SLE patients has been reported [7,8]. In a study by Vasdaz et al., which included 12 patients with LN, an increased renal expression of NRP-1 was observed in deposits located exclusively in damaged glomerular areas, correlating with proteinuria and the chronicity index (indicative of endothelial damage) [9,10]. Additionally, a study by Torres-Sanchez et al. found that urinary NRP-1 mRNA levels were significantly higher in patients with active LN compared to those with active SLE without renal involvement. Similar results were observed at the tissue level. Responders had higher NRP-1 expression than non-responders. Immunohistochemical analysis of renal biopsies revealed that NRP-1 staining was primarily present in the glomeruli and, to a lesser extent, in the tubules. Within the glomeruli, NRP-1 was mainly expressed along the endothelial cell membrane and slightly in mesangial cells [11].

The role of NRP-1 in renal recovery remains a subject of debate. Some studies suggest that NRP-1 worsens acute kidney injury by promoting fibrosis through the downregulation of Cox4il enzyme crotonylation and activation of Smad3 [12]. Additionally, NRP-1 has been associated with acute renal failure by driving the accumulation of miR-21a-3p in renal tubular epithelial cells, which leads to metabolic dysfunction [13]. In contrast, in type 1 diabetes, NRP-1 plays a different role by mediating immune suppression and regulation via the activation of regulatory T cells (Tregs) [14]. In vitro experiments with human primary renal cells suggest that NRP-1 may have a dual role in renal recovery, enhancing angiogenesis, and promoting mesangial regeneration [11].

Given this background, it is important to determine whether NRP-1 levels are directly associated with histological renal improvement. In human studies, performing serial renal biopsies is challenging and raises serious ethical concerns. Therefore, the primary objective of this study is to longitudinally evaluate NRP-1 protein levels in urine, serum, and renal tissue in an NZB/W F1 experimental LN model treated with cyclophosphamide. The goal is to establish whether NRP-1 could serve as a reliable biomarker for monitoring renal recovery in LN by demonstrating a direct correlation between its levels and renal tissue recovery.

## 2. Results

### 2.1. Cyclophosphamide Treatment Delays Disease Onset in NZB/W F1 Mice

Two groups of 15 female NZB/W F1 mice were treated with either CYC (50 mg/kg) or an equivalent volume of PBS at weeks 24 and 28. Survival analysis revealed that the PBS control group experienced earlier mortality compared to the CYC-treated group (Figure 1A). While CYC-treated mice began to die at week 36 (83% survival at 8.5 months), 66% of the PBS control group had already died by week 34. By week 38, survival rates declined to 16% in the PBS group and 66% in the treatment group. At the study endpoint (week 42), survival in the control group was 0%, whereas the CYC-treated group maintained a survival rate of 50%.

CYC treatment also resulted in a gradual reduction in serum levels of both total immunoglobulin (IgG) and IgG anti-dsDNA antibodies over time. At weeks 26 and 32, IgG levels in the treated group were significantly lower than those in the control group (week 26: 119.6 IU/mL vs. 180.4 IU/mL; week 32: 133.4 IU/mL vs. 182.6 IU/mL; *p* < 0.05). Similarly, anti-dsDNA antibody levels were significantly decreased in the CYC-treated group at these time points (week 26: 11.5 IU/mL vs. 18.5 IU/mL; week 32: 16.0 IU/mL vs. 25.5 IU/mL; *p* < 0.05).

Furthermore, CYC treatment delayed the onset of proteinuria and reduced the protein/creatinine ratio at weeks 26 (114 µg/mg vs. 919.1 µg/mg), 32 (904 µg/mg vs. 2324 µg/mg), and at the endpoint (973 µg/mg vs. 2732 µg/mg) (*p* < 0.05) (Figure 1B).

Histologically, the CYC-treated mice exhibited significantly reduced mesangial proliferation from week 32 through to the study endpoint (Figure 2A,B). However, no statistically significant differences were observed in the activity and chronicity indices between the two groups (Appendix A). Additionally, immunofluorescence analysis revealed the deposition of complement component C3 and IgG (Figure 2C,D). In the CYC-treated mice, a time-dependent reduction in both C3 and IgG levels was noted, with statistically significant reductions in IgG at week 34 and in both at the study endpoint (Figure 2C,D).

### 2.2. NRP-1 Expression Is Increased in the Kidneys of Treated NZB/W F1 Mice

Previous research has shown that NRP-1 gene expression is elevated in biopsies from patients with LN [10,11] and in renal pericytes within the glomeruli of mouse models [12]. Our study demonstrated that renal NRP-1 expression, as assessed by immunofluorescence, increased in the treated mice from week 26 onward when compared to the control group. Analysis of expression levels prior to treatment at week 20 revealed no significant differences between the groups (1.4-fold change in the treated group vs. 1.1-fold change in the control group, *p* < 0.05, Figure 3A). Additionally, NRP-1 expression was predominantly localized in the tubular region of the kidney, although it was also present in the glomerular area (Figure 3A).

In serum, no significant increase in NRP-1 expression was observed in the CYC-treated mice compared to the untreated mice at any study week, and although a quantitative increase was noted at week 26, it did not reach statistical significance (Appendix A). In urine, a significant elevation in NRP-1 levels was observed in the CYC-treated mice at all study weeks, except week 20 (*p* < 0.05). At week 32, NRP-1 levels were substantially higher in the treated group (11.58 ng/mL vs. 108.8 ng/mL, Figure 3B).

Longitudinal analysis revealed distinct patterns in NRP-1 levels between the CYC-treated and non-treated groups. In the non-treated group, serum NRP-1 levels remained consistently higher compared to in urine throughout the progression of lupus nephritis. Conversely, in the CYC-treated group, serum NRP-1 levels were higher than in urine until week 26 (46.14 ng/mL in serum vs. 16.01 ng/mL in urine at week 26). However, by week 32, a significant shift occurred, with higher NRP-1 levels in urine (*p* = 0.005), suggesting renal production. At the study endpoint, NRP-1 levels in serum and urine were comparable. Within the treated group, serum peak NRP-1 expression occurred at week 26, while in urine, the peak was observed at week 32 (Figure 3B).

Overall, no differences in NRP-1 expression were detected in tissue or serum during the early stages of the study. However, the longitudinal analysis indicated that urinary NRP-1 (uNRP-1) levels increased significantly with disease progression and treatment, suggesting that uNRP-1 could serve as a valuable marker for monitoring renal recovery.

### 2.3. uNRP-1 Expression Inversely Correlates with Disease Biomarkers, Indicating a Role in Renal Recovery

A significant negative correlation was observed between uNRP-1 levels and key disease activity biomarkers, including IgG, anti-dsDNA, and the protein/creatinine ratio. Specifically, higher levels of IgG (r = −0.46, *p* = 0.0004), anti-dsDNA (r = −0.28, *p* = 0.03), and the protein/creatinine ratio (r = −0.27, *p* = 0.04) were associated with lower levels of NRP-1 (Figure 4). Additionally, a statistically significant negative correlation between mesangial proliferation and uNRP-1 levels was identified in the histological analysis (r = −0.563, *p* = 0.009), (Figure 4). Together, these findings, along with the clinical parameters, support the potential involvement of NRP-1 in renal tissue recovery in LN.

### 2.4. ROC Curve Analysis of uNRP-1 Levels for Predicting Renal Recovery in NZB/W F1 Mice

We evaluated renal samples from the CYC-treated and non-treated NZB/W F1 mice by examining mesangial proliferation, classifying them into renal recovery and non-recovery groups. Tissues with mesangial proliferation of less than 40% were classified as indicative of renal recovery. Based on this criterion, 73% of the CYC-treated mice (11 out of 15) were identified as experiencing renal recovery, while none of the non-treated mice (0 out of 15) showed signs of recovery. As a result, 11 mice were included in the renal recovery group and 19 in the non-recovery group. Next, we investigated the predictive capacity of uNRP-1 for renal recovery using ROC curve analysis (Figure 5A). The area under the curve (AUC) of 0.890 describes a statistically significant and robust ability for uNRP-1 to differentiate renal recovery from non-recovery, (*p* = 0.001, Figure 5A). Using the ROC curve, a uNRP-1 cut-off value of 34.40 ng/mL was established for predicting favorable renal recovery, demonstrating 100% sensitivity and 88% specificity (Figure 5A). Longitudinal analysis utilizing this cut-off demonstrated its effectiveness in distinguishing between the two groups throughout the study period (Figure 5B). These findings suggest that uNRP-1 levels could serve as a valuable biomarker for assessing renal recovery, correlating with improvements in mesangial proliferation in the kidney.

## 3. Discussion

Despite advances in the diagnosis and treatment of SLE in recent decades, LN remains one of the most severe and morbidity-prone complications [15]. In our study, we observed significantly elevated levels of NRP-1 in both urine and renal tissue of NZB/WF1 mice with LN that received CYC. Moreover, histological correlations revealed a negative association between uNRP-1 and both mesangial proliferation and disease activity markers (anti-dsDNA, IgG, and protein/creatinine ratio). The ROC curve analysis for uNRP-1 demonstrated an AUC of 0.890, indicating its excellent capacity to predict renal recovery.

Currently, therapeutic response in LN is monitored clinically, with improvements in proteinuria, urinary sediment, and renal function serving as indicators of renal inflammation resolution. However, there is often a discrepancy between clinical findings and the actual status of the renal parenchyma, making serial biopsies the ideal method for assessment, though they are clinically impractical in many cases [16,17]. The “gold standard” for diagnosing LN remains renal biopsy, which provides, at the time of the procedure, detailed information about the type and severity of renal tissue damage as well as data of histological improvement following treatment. However, this method is invasive, costly, and may be inaccessible in certain healthcare systems [3]. To date, only proteinuria is a well-established marker for diagnosis, disease activity, renal response, and flares [18] since current serological biomarkers lack sensitivity and specificity to predict renal outcomes. Therefore, new biomarkers that reflect the underlying renal inflammatory process and better predict LN progression and treatment response are needed. Urinary biomarkers are particularly relevant as they can be measured non-invasively and may better reflect the compartmentalized renal response in LN, unlike serum studies that are non-specific to the kidney. Several proteins, such as adhesion molecules, autoantibodies, chemokines, complement proteins, and cytokines, have been recognized as possible biomarkers of disease activity in cross-sectional studies of LN patients, but they have not been validated in larger cohorts. Whereas many of them have been reported to be elevated in LN compared to healthy controls, not all of them are able to distinguish patients with active LN (ALN) from inactive SLE or correlate with early clinical response [19]. Recently, urine proteomic studies have identified new urinary biomarkers associated with monocyte/neutrophil degranulation (PR3, S100A8, azurocidin, catalase, cathepsins, MMP8), macrophage activation (CD163, CD206, galectin-1), wound healing/matrix degradation (nidogen-1, decorin), and IL-16 in patients with active LN [19] that correlated with histological activity and treatment response.

CYC treatment in LN experimental models has been shown to reduce glomerular proliferation and levels of anti-dsDNA and IgGs while increasing glomerulosclerosis as the disease progresses until model death [20]. Our study found that NZB/W F1 mice with active LN treated with CYC exhibited improved survival compared to untreated mice, with a survival rate of 52%, consistent with other studies [20,21]. Histological analysis revealed a statistically significant reduction in mesangial proliferation in the CYC-treated mice. Less renal deposition of immune complexes was also noted, as indicated by decreased levels of C3 and IgG in kidney tissue. However, no significant differences were observed in parameters related to the glomerular index, including glomerular sclerosis, fibrous crescents, interstitial fibrosis, and tubular atrophy. This discrepancy may be due to the lower CYC dose used in our study compared to other research, which could have contributed to the lack of glomerulosclerotic development.

Several studies have demonstrated that high titers of anti-dsDNA antibodies and low levels of C3 and C4 often precede an LN flare [22]. However, not all patients with elevated anti-dsDNA antibody titers develop nephritis [23]. Similarly, changes in complement levels have shown variable results in predicting kidney flares or response to therapy [24]. In contrast, proteinuria is a well-established marker for diagnosis, disease activity, renal response, and flares [25]. In our study, we found a negative correlation between NRP-1 and traditional serum biomarkers such as anti-dsDNA, IgG, and the protein/creatinine ratio. Histological examination revealed decreased mesangial proliferation and immune complex deposition in the treatment group, which exhibited higher NRP-1 levels. This suggests a potential role of NRP-1 at the mesangial level. Additionally, our study found that uNRP-1 was negatively correlated with mesangial proliferation.

NRP-1 has been found to be elevated in both histological and urinary samples from LN patients and correlated with renal response [11]. Consistent with previous findings, longitudinal analysis of serum and urinary NRP-1 levels revealed no significant increase in serum NRP-1 expression in the CYC-treated mice compared to the non-treated mice. In contrast, a significant increase in uNRP-1 levels was observed in the CYC-treated mice throughout the study, with the most pronounced rise occurring at week 32. Immunofluorescence analysis of kidney tissue demonstrated an increase in NRP-1 in the CYC-treated mice starting at week 26. The lack of any change in serum NRP-1 levels and the significant differences at the urinary and tissue level may suggest renal NRP-1 production during recovery. uNRP-1 levels discriminate recovery from non-recovery with a sensitivity and specificity of 100% and 88%. We determined a uNRP-1 cutoff that accurately discriminated mice in remission, suggesting that its measurement could serve as a sensitive indicator of early renal response in LN. This study also demonstrated a negative correlation between NRP-1 levels and traditional serum disease biomarkers, including anti-dsDNA, IgG, and the protein/creatinine ratio. Histological analysis further revealed reduced mesangial proliferation and immune complex deposition in the treatment group with elevated NRP-1 levels, suggesting a potential role for NRP-1 at the mesangial level.

Microarray analysis of NZB/W F1 mice with lupus nephritis at different stages indicates that hypoxia and metabolic stress are key pathways involved in renal remission, more so than the production of renal chemokines and cytokines [26,27]. NRP-1 enhances VEGF activity, thereby intensifying its protective effects through the promotion of angiogenesis [11]. The proliferation of new blood vessels and the resulting increase in oxygen supply to the glomerular endothelium create an environment conducive to renal recovery [28]. NRP-1 is directly associated with angiogenesis and hypoxia, further supporting its potential role in renal recovery. In addition, in vitro studies suggest that NRP-1 can promote renal recovery through endothelial proliferation and migration, mesangial migration, and local T cell cytotoxicity [11].

Thus, our findings indicate that kidney and urine NRP-1 levels are closely related to recovery from lupus nephritis. uNRP-1 is a promising candidate for evaluation as an early marker of renal response, potentially reducing the need for renal biopsies to assess therapeutic response. However, the use of murine models and murine renal tissue in this study presents a significant limitation, as the pathophysiological mechanisms underlying lupus nephritis may differ between mice and humans. While murine models provide valuable insights into disease processes, the translational relevance of these findings may be constrained due to inherent biological differences. Consequently, further studies are required to validate these findings and explore the potential of uNRP-1 as a biomarker in clinical practice.

## 4. Materials and Methods

### 4.1. Animal Models

Thirty female NZBWF1/J mice were purchased from The Jackson Laboratory (Bar Harbor, ME, USA) at 15 weeks of age (strain: #100008, identification code: RRID:IMSR_JAX:100008). This hybrid strain develops an autoimmune disease resembling human SLE, characterized by elevated antinuclear antibodies, hemolytic anemia, proteinuria, and progressive glomerulonephritis due to immune complexes. Proteinuria and renal disease typically begin around weeks 22–24. Other animal models were considered for this study, such as MRL/lpr and BXSB/Yaa [29]. MRL/lpr mice produce autoantibodies against dsDNA and Sm, resulting in the formation of large amounts of immunocomplexes that induce renal and skin pathology. BXSB/Yaa mice develop a lupus-like disease characterized by lymphoid hyperplasia, immune-complex-mediated nephritis, antinuclear antibodies (ANA), and elevated serum titers of the retroviral glycoprotein gp70. While both strains could serve as viable models for lupus nephritis research, the NZB/W F1 model is preferred due to its gradual disease progression, which allows us to observe the potential role of NRP-1 in lupus nephritis progression. Furthermore, the genetic and immunological features of NZB/W F1 mice are well documented and closely resemble those of human lupus nephritis, providing greater translational relevance to our findings [29].

The mice were divided into two treatment groups (*n* = 15 each): one group received cyclophosphamide treatment, while the control group received saline solution. Pharmacological administration was performed intravenously via the tail at two-week intervals, specifically at weeks 24 and 28. The control group received saline solution, and the CYC group received an IV dose of 50 mg/kg per administration in 20 mL volume/mouse of CYC (Cytoxan) (Appendix A).

Subgroups of mice were sacrificed at weeks 20, 26, and 32 to obtain longitudinal kidney samples (*n* = 3 per week per group). Blood and urine samples were also collected at the time of sacrifice. The remaining mice were sacrificed either upon meeting specific criteria or at the study endpoint (week 43). Since the last mouse in the control group died at week 42, the study endpoint was set at week 43 to ensure sample collection occurred at a comparable time point for accurate comparisons. Criteria for early termination included signs of pain or distress, such as hair loss, a 20% sudden reduction in body weight, curvature, or atypical behaviors. All procedures adhered to the 3Rs principles for animal experimentation and were approved by the Animal Experimentation Committee of the Generalitat of Catalonia (project reference number 11446). Blood and urine samples were also collected from non-sacrificed mice at weeks 20, 26, and 32 to quantify NRP-1 protein levels, anti-dsDNA, IgG, and proteinuria throughout the disease progression.

### 4.2. Histological Analysis of Renal Tissue

Kidney samples from NZB/W F1 mice were collected at weeks 20, 26, 32 (*n* = 3 at each time point), and at the study endpoint (*n* = 6) following euthanasia. Tissues were carefully dissected to preserve integrity, fixed in paraformaldehyde, dehydrated through a series of alcohol gradients, and embedded in paraffin. Sections (approximately 5 µm thick) were cut using a microtome, stained with hematoxylin and eosin, and subjected to immunofluorescence staining. Evaluation was conducted by the research team from the SLE Unit and pathology experts at Vall d’Hebron University Hospital. Assessment included activity indices, chronicity, and mesangial proliferation within the renal tissue, using a scoring system specifically designed for murine models [30].

### 4.3. Immunofluorescence for IgG, C3, or NRP-1 Protein Levels in Kidney Tissue

Five-micrometer sections of renal tissue were deparaffinized with xylene and rehydrated through a descending series of alcohol concentrations (99% to 70% ethanol). Antigen retrieval was performed using proteinase K. Non-specific binding sites were blocked with 5% bovine serum albumin (BSA, Irving, TX, USA) for 1 h at room temperature. Primary rabbit anti-mouse antibody specific for IgG (GTX26709, GeneTex, Irvine, CA, USA), C3 (GTX101316, GeneTex, Irvine, CA, USA), or NRP-1 (sc-5307, Santa Cruz Biotechnology, Dallas, TX, USA) was applied at a dilution of 1:250, 1:250, or 1:50, respectively, and incubated overnight at 4 °C. After incubation, primary antibodies were removed by several washes with phosphate-buffered saline (PBS, Arlington, VA, USA) containing 0.25% of Triton-X100. A secondary goat anti-rabbit antibody conjugated with DyLight488 (diluted at 1:250, GTX213110-04, GeneTex, Irvine, CA, USA) was applied and incubated for 2 h at room temperature. Unbound secondary antibodies were washed away with PBS containing 0.25% of Triton-X100, and cellular nuclei were stained with 4′, 6-diamidino-2-phenylindole (DAPI). The tissue was examined using an Olympus BX61 fluorescence microscope, and fluorescence intensity levels were quantified using the ImageJ Fiji software, version 1.45 (more details in the Appendix A).

### 4.4. Quantification of NRP-1 Levels in Blood and Urine Samples

NRP-1 levels in blood and urine were analyzed using an enzyme-linked immunosorbent assay (ELISA). Blood and urine samples were collected at weeks 20, 26, 32, and at the study endpoint before euthanasia. The samples were processed to obtain serum and stored at−80 °C. Neuropilin-1 ELISA Kits (Fine Biotech, Wuhan, China) were used for the analysis. Plates were coated with antibodies specific for NRP-1, and a blocking solution was applied to prevent non-specific protein binding. Diluted serum and urine samples, along with controls of a known NRP-1 concentration, were added to the wells of the ELISA plate. The plate was incubated to allow the NRP-1 to bind to the coated antibodies. After incubation, the plate was washed, and secondary antibodies conjugated with streptavidin were added. A chromogenic substrate was then introduced, producing a color change proportional to the NRP-1 concentration. Absorbance was measured at 450 nm using an ELISA plate reader. The absorbance values of the samples were compared with those of the controls to determine the NRP-1 concentration in the serum and urine.

### 4.5. Determination of Anti-dsDNA and IgG Levels in Serum

Quantifications of anti-dsDNA and IgG antibody in NZB/WF1 mice were performed at weeks 20, 26, 32, and at the study endpoint using the ELISA technique. For the anti-dsDNA analysis, we followed the manufacturer’s instructions for the kit “anti-double stranded DNA, dsDNA IgG, ELISA” (MBS7225215, MyBiosource, Vancouver, Canada). The “Mouse IgG (Total) Uncoated ELISA Kit with Plates” was used for quantifying IgG levels (ThermoFisher, Carlsbad, CA, USA). ELISA plates were coated overnight at 4 °C with either dsDNA or anti-mouse IgG antibodies in PBS. After coating, the plates were washed and blocked with 5% BSA in PBS for 2 h at room temperature. Serum samples diluted in a specific buffer containing PBS and BSA were added to the ELISA plates and incubated overnight at 4 °C. After incubation, the plates were washed, and secondary antibodies conjugated with horseradish peroxidase (HRP) were added for detection. A chromogenic substrate (TMB, 3, 3′, 5, 5′-tetramethylbenzidine) was added, and the reaction was stopped with sulfuric acid. Optical density (OD) was measured at 450 nm using an ELISA plate reader. OD readings from the samples were compared with those of positive and negative controls to determine the relative concentration of anti-dsDNA and IgG antibodies.

### 4.6. Determination of Urine Protein/Creatinine Ratio

To determine the protein/creatinine ratio in the urine of NZB/WF1 mice, urine samples were manually collected at weeks 20, 26, 32, and at the study’s endpoint. The samples were stored at−20 °C until analysis. We measured total protein and creatinine levels using a QuantiChrom Protein Creatinine Ratio Assay Kit (BioAssay Systems, Hayward, CA, USA, #DPCR-100), following the manufacturer’s instructions. Optical density values from the samples, internal standards, and blank wells were used to calculate protein and creatinine concentrations. The protein/creatinine ratio was then determined by dividing the protein concentration (µg/dL) by the creatinine concentration (mg/dL), resulting in a ratio expressed as µg of protein per mg of creatinine. This ratio provides an accurate assessment of kidney function and disease progression.

### 4.7. Statistical Analysis

Data are presented as mean ± standard deviation (SD). Statistical analysis was performed using Student’s *t*-test or one- or two-way ANOVA with Prism GraphPad, version 7.0 (GraphPad Software, v 7.0, San Diego, CA). For survival analysis, animals sacrificed to obtain longitudinal kidney samples were not included, and the analysis was conducted up to week 42. Correlation between two parameters was analyzed using Spearman’s rank-order correlation. Receiver operating characteristic (ROC) curves were used to assess the predictive value, sensitivity, and specificity of each biomarker, with appropriate cut-off points selected according to the Youden index. The optimal cut-off point is identified as the point on the ROC curve closest to the top-left corner. *p*-values <0.05 were considered statistically significant.

## Figures and Tables

**Figure 1 ijms-25-11364-f001:**
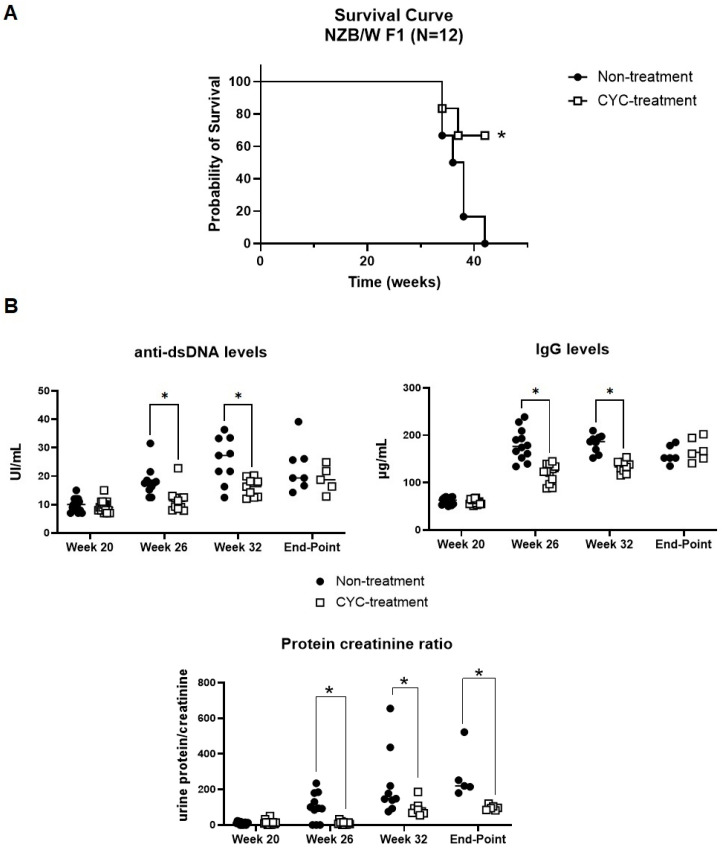
The ameliorative effect of CYC on disease progression in NZB/W F1 mice. (**A**) Kaplan–Meier survival curve for NZB/W F1 mice treated with cyclophosphamide or saline (CYC-treated and non-treated groups, respectively). The log-rank test was used for survival analysis, with *p*-values ≤ 0.05 considered statistically significant. (**B**) Serum and urine biomarkers were measured to assess the progression of LN in both treated and non-treated mouse groups. Serum autoantibodies to double-stranded DNA (anti-dsDNA) and immunoglobulin (IgG) titers, as well as the urine protein/creatinine ratio (µg/mg), were determined using enzyme-linked immunosorbent assay (ELISA). One-way ANOVA followed by Bonferroni’s test and Student’s *t*-test were used to compare biomarker concentrations between the non-treated and CYC-treated groups. * *p* < 0.05.

**Figure 2 ijms-25-11364-f002:**
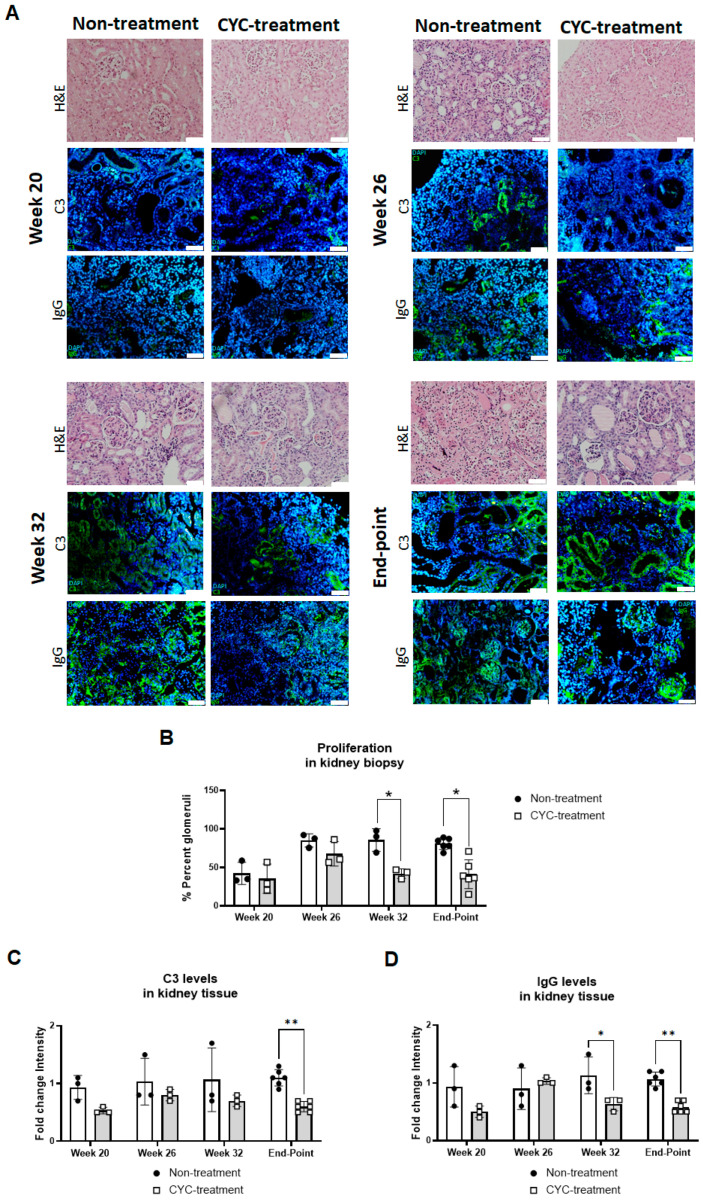
Analysis of kidney tissue from untreated and CYC-treated groups. (**A**) Renal tissue was subjected to hematoxylin and eosin staining as well as immunofluorescence analysis for IgG and complement component C3. Cell nuclei were stained with DAPI (blue), while IgG and C3 proteins were labeled with FITC (green). Scale bar = 50 µm. Average scores for evaluation were obtained from the Vall d’Hebron pathology group to assess the percentage of glomeruli (**B**) and the staining intensity for IgG (**C**) and C3 (**D**) in renal tissue. One-way ANOVA followed by Bonferroni’s test and Student’s *t*-test were used to compare biomarker concentrations between groups. * *p* < 0.05, ** *p* < 0.01.

**Figure 3 ijms-25-11364-f003:**
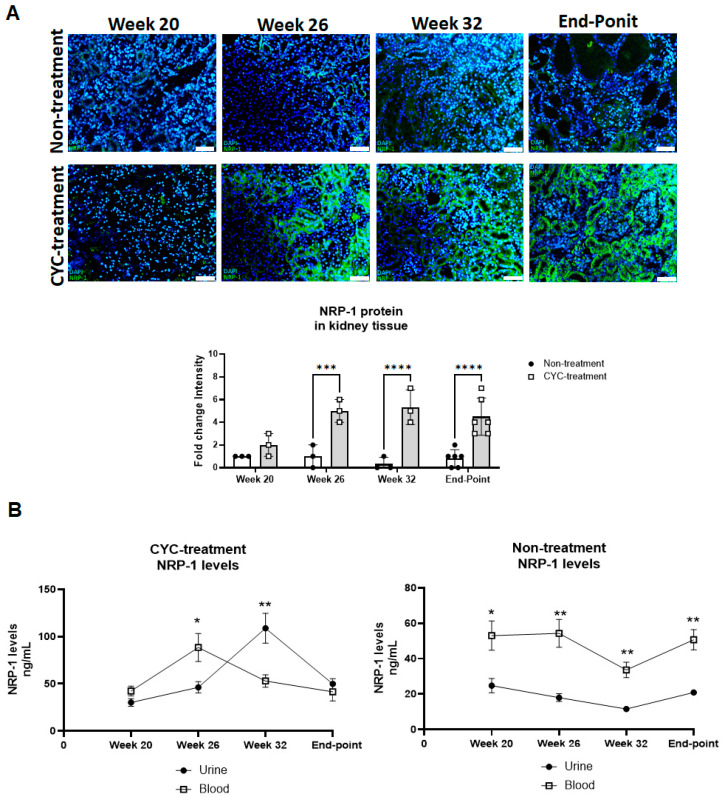
Longitudinal analysis of NRP-1 protein levels during nephritis progression in NZB/W F1 mice. (**A**) Immunofluorescence staining for NRP-1 was performed on kidney tissue samples from both non-treated and CYC-treated groups at weeks 20, 26, 32, and at the study endpoint. Cell nuclei were stained with DAPI (blue), and NRP-1 was labeled with FITC (green). Scale bar = 50 µm. The intensity of NRP-1 immunofluorescence was quantified using the ImageJ Fiji software (version 1.45). Statistical comparisons between groups were performed using Student’s *t*-test. *** *p* < 0.0001, **** *p* < 0.00001. (**B**) Serum and urine levels of NRP-1 were measured at weeks 20, 26, 32, and at the study endpoint in both CYC-treated and non-treated NZB/W F1 mice groups. One-way ANOVA followed by Bonferroni’s post hoc test and Student’s *t*-test were used to compare NRP-1 levels between urine and serum at each time point. * *p* < 0.05, ** *p* < 0.001.

**Figure 4 ijms-25-11364-f004:**
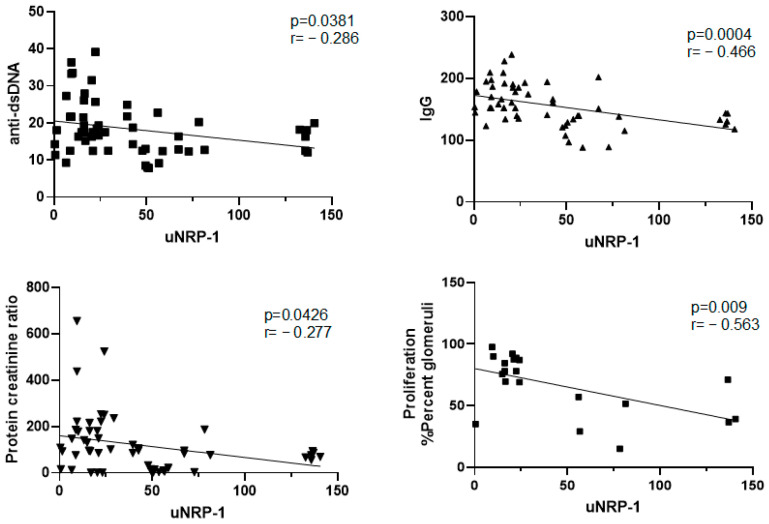
Correlation between urinary NRP-1 levels (uNRP-1) and anti-dsDNA and IgG levels, protein/creatinine ratio, and mesangial proliferation. Spearman’s rank-order correlation was utilized to assess the relationships between these parameters, with significance levels and correlation coefficients indicated in each graph.

**Figure 5 ijms-25-11364-f005:**
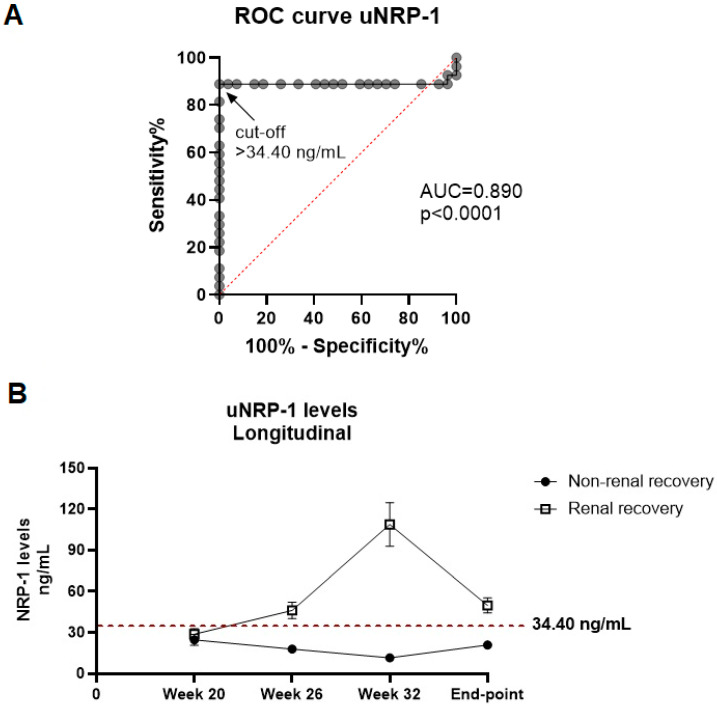
Urinary NRP-1 levels as a potential biomarker for monitoring renal recovery. (**A**) Receiver operating characteristic (ROC) curve analysis of uNRP-1 was conducted using an optimal binary logistic regression model, combining data from both cohorts, to distinguish between renal recovery and non-recovery in NZB/W F1 mice (*n* = 11 for renal recovery, *n* = 19 for non-recovery). The area under the ROC curve (AUC) is provided. (**B**) Longitudinal analysis of uNRP-1 levels during the progression of lupus nephritis in NZB/W F1 mice was performed, categorizing mice based on whether they achieved renal recovery. Renal recovery was determined through histological analysis of kidney tissue, with mice exhibiting more than 40% mesangial proliferation being classified into the non-recovery group. Throughout the study period, mice in the renal recovery group consistently maintained uNRP-1 levels above 34.40 ng/mL.

## Data Availability

All data in this study are available from the corresponding author upon reasonable request.

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
