# Peer review of "Neuropilin-1 as a Key Molecule for Renal Recovery in Lupus Nephritis: Insights from an NZB/W F1 Mouse Model"

_ijms, 2024, doi:10.3390/ijms252111364_

Round 1
Reviewer 1 Report
Comments and Suggestions for Authors
I appreciate the opportunity to review this manuscript. The topic is relevant to the field, suggesting the correlation of urinary NRP-1 levels with renal histological recovery, reinforcing its potential as a biomarker for renal response in lupus nephritis. However, there are several major areas that need clarification and improvement as follows.
1. My major concern is regarding the survival analysis performed by the authors. According to the methods section (line 321 and Figure S2), the mice were sacrificed, which raises questions about the validity of the survival analysis. Please clarify this discrepancy or provide a rationale for this approach.
2. The description in line 320 is unclear and requires readers to perform backcalculation. To enhance clarity, I suggest rephrasing it to directly state the dosing as: "an IV dose of 50 mg/kg per administration in ... μL volume/mouse."
3. Please provide a reference for a line no. 338-340.
4. The manuscript lacks detailed information on the methodology for ImageJ analysis. Since different proteins have varying expressions and fluorescence, the analysis can differ. Please include detailed steps of the ImageJ analysis to ensure reproducibility and transparency. It can be added in a supplemental material.
5. For section 4.5, provide the name of a kit.
6. Please discuss why the NZBW/F1 mouse model was chosen for this study over other models, such as SLE or MRL/lpr mice.
Comments on the Quality of English Language1. Figure 1 legend: please check if you want to change "lupus nephritis " to "disease progression" (line no. 101) as apart from urinary marker, serum and survival markers are also present in this figure.
2. Ensure that abbreviations are defined upon their first mention in the abstract, main text, and figure legends. After the first mention, please use abbreviations consistently throughout the manuscript.
3. Please add a regression line in Figure 4 to clearly show the trend in the data and improve interpretability.
Author Response
I appreciate the opportunity to review this manuscript. The topic is relevant to the field, suggesting the correlation of urinary NRP-1 levels with renal histological recovery, reinforcing its potential as a biomarker for renal response in lupus nephritis. However, there are several major areas that need clarification and improvement as follows.
- My major concern is regarding the survival analysis performed by the authors. According to the methods section (line 321 and Figure S2), the mice were sacrificed, which raises questions about the validity of the survival analysis. Please clarify this discrepancy or provide a rationale for this approach.
We acknowledge the concern regarding the survival analysis. To clarify, a subgroup of animals was euthanized at weeks 20, 26, and 32 to obtain longitudinal kidney samples during the study, and these animals were excluded from the survival analysis. No animals in the PBS-treated group survived beyond week 42. In the CYC-treated group, four animals survived past week 42. To ensure consistency in sample collection across groups, these CYC-treated animals were euthanized at week 43 for kidney sample collection. Consequently, the survival analysis was conducted up to week 42. This rationale has been clarified in the Methods section (Page 12, Line 339-348, and Page 14, Line 424-426) and in Figure S2.
- The description in line 320 is unclear and requires readers to perform back calculation. To enhance clarity, I suggest rephrasing it to directly state the dosing as: "an IV dose of 50 mg/kg per administration in ... μL volume/mouse."
Thank you for your suggestion. The sentence has been rephrased according to your recommendation to make it clearer for the readers as follows in line no. 328-329: “The control group received saline solution, and the cyclophosphamide group received an IV dose of 50mg/kg per administration in 20mL volume/mouse of cyclophosphamide (Cytoxan).”
- Please provide a reference for a line no. 338-340.
I has been provided in the manuscript as reference number 30 (Page 12 , line 360).
- The manuscript lacks detailed information on the methodology for ImageJ analysis. Since different proteins have varying expressions and fluorescence, the analysis can differ. Please include detailed steps of the ImageJ analysis to ensure reproducibility and transparency. It can be added in a supplemental material.
We have included a detailed methodology for ImageJ analysis in the supplemental materials and cited it in the main manuscript (Page 13, Line 377-378).
- For section 4.5, provide the name of a kit.
Following your suggestion, we have included the name of the kit in Section 4.5 (Page 13, line 396-398).
- Please discuss why the NZBW/F1 mouse model was chosen for this study over other models, such as SLE or MRL/lpr mice.
We have included a discussion about animal model in Lupus Nephritis in the methodology after a brief description of the current NZBW/F1 animal model (Page 12, line 318-332).
Comments on the Quality of English Language
- Figure 1 legend: please check if you want to change "lupus nephritis " to "disease progression" (line no. 101) as apart from urinary marker, serum and survival markers are also present in this figure.
Thank you for the recommendation. We have updated the figure legend (Page 4, Line 109-117) to reflect the inclusion of systemic blood markers, alongside renal disease markers: 'The ameliorative effect of cyclophosphamide on disease progression in NZB/W F1 mice.
- Ensure that abbreviations are defined upon their first mention in the abstract, main text, and figure legends. After the first mention, please use abbreviations consistently throughout the manuscript.
Thank you for the recommendation; we have reviewed the entire text accordingly.
- Please add a regression line in Figure 4 to clearly show the trend in the data and improve interpretability.
Following your suggestion, the regression line has been added to Figure 4 (Page 8).

Reviewer 2 Report
Comments and Suggestions for Authors
In this paper, the authors investigated the role of Neuropilin-1 (NRP-1) as a potential biomarker for renal recovery, using the NZB/W F1 mouse model. The study included a detailed analysis of NRP-1 levels in urine, serum and renal tissue, highlighting its potential as a predictive marker of renal recovery. The design of the study, the experimental model chosen, and the focus on NRP-1 as a biomarker provide interesting and potentially valuable insights, but there are still issues in the manuscript that need to be addressed. It is therefore recommended that it be published after a major revision, and that the following points be carefully addressed.
1. The abstract is confusing. The authors should have explicitly described the innovation of this study.
2. The introduction provides an overview of the function of NRP-1, but does not adequately discuss its relation to renal recovery. Authors should add the latest research results.
3. The authors' study found no statistically significant differences in blood NRP-1 levels. However, progressive increases in renal and urinary NRP-1 expression were observed in the treated group. The authors need to explain why.
4. The sample size of the subgroups of mice (n=3) seems relatively small, especially for histological evaluations.
5. What does “ug/mg” represent on the y-axis of Figure 1?
6. How was the cutoff of 34.40 ng/mL obtained?
7. There are significant differences between the animal models in this study and human kidney disease. The authors should have clearly stated the limitations of this study.
8. There is no detailed experimental procedure for immunofluorescence experiments.
9. Editing was required to improve the English throughout the manuscript.
Comments on the Quality of English LanguageModerate editing of English language required.
Author Response
In this paper, the authors investigated the role of Neuropilin-1 (NRP-1) as a potential biomarker for renal recovery, using the NZB/W F1 mouse model. The study included a detailed analysis of NRP-1 levels in urine, serum and renal tissue, highlighting its potential as a predictive marker of renal recovery. The design of the study, the experimental model chosen, and the focus on NRP-1 as a biomarker provide interesting and potentially valuable insights, but there are still issues in the manuscript that need to be addressed. It is therefore recommended that it be published after a major revision, and that the following points be carefully addressed.
- The abstract is confusing. The authors should have explicitly described the innovation of this study.
To address the concerns regarding the clarity of the abstract, we have revised it to explicitly highlight the innovations of this study (Page 1, lines 14-30).
- The introduction provides an overview of the function of NRP-1, but does not adequately discuss its relation to renal recovery. Authors should add the latest research results.
According to your recommendation, we specified the latest findings in the literature regarding the relationship between NRP-1 and renal recovery (Page 2, lines 70-78). We added the following bibliography:
- Li Y, Wang Z, Xu H, et al. Targeting the transmembrane cytokine co-receptor neuropilin-1 in distal tubules improves renal injury and fibrosis. Nat Commun. 2024;15:5731.
- Zou Z, Lin Q, Yang H, Liu Z, Zheng S. Nrp-1 mediated plasmatic Ago2 binding miR-21a-3p internalization: a novel mechanism for miR-21a-3p accumulation in renal tubular epithelial cells during sepsis. Biomed Res Int. 2020 Aug 18;2020:2370253.
- Khamechian T, Irandoust B, Mohammadi H, Nikoueinejad H, Akbari H. Association of regulatory T cells with diabetes type-1 and its renal and vascular complications based on the expression of Forkhead Box Protein P3 (FoxP3), Helios and Neuropilin-1. Iran J Allergy Asthma Immunol. 2018 Apr;17(2):151-157.
- Torres-Salido MT, Sanchis M, Solé C, et al. Urinary neuropilin-1: A predictive biomarker for renal outcome in lupus nephritis. Int J Mol Sci. 2019, 20, 4601.
- The authors' study found no statistically significant differences in blood NRP-1 levels. However, progressive increases in renal and urinary NRP-1 expression were observed in the treated group. The authors need to explain why.
In serum samples, no significant increase in NRP-1 expression was observed in CYC-treated mice compared to untreated mice at any study week. However, significant differences were found in urine and kidney staining, with the most pronounced increase occurring at week 32. To illustrate this difference more clearly, we included Figure S2, which shows urinary NRP-1 levels between CYC-treated and untreated mice at study time points (Page 7, lines 156-158).
These results suggest that NRP-1 production associated with renal outcomes may occur at the renal level. Given the close relationship between urine and kidney function, the elevated NRP-1 levels in urine reflect localized production. In contrast, the absence of changes in serum NRP-1 levels between the two groups indicates that NRP-1 production related to kidney improvement may be organ-dependent. We have discussed this in main manuscript (Page 11, lines 278-294).
- The sample size of the subgroups of mice (n=3) seems relatively small, especially for histological evaluations.
In line with the 3Rs principles for animal experimentation, we aimed to minimize the number of animals used in the study. While we acknowledge that the sample size for histological evaluations is small, we conducted five replicates for each analysis. These were evaluated by research teams from the SLE Unit and pathology experts at Vall d'Hebron University Hospital to ensure consistency. We have included a detailed methodology for this analysis in the supplementary information (Page 13, line 378).
- What does “ug/mg” represent on the y-axis of Figure 1?
It was a typographical error; it has been corrected to 'urine protein/creatinine” (Figure 1, Page 4).
- How was the cutoff of 34.40 ng/mL obtained?
We determined the cut-off for urinary NRP-1 levels (uNRP-1) using the ROC curve generated from an optimal binary logistic regression model that combined data from both cohorts. This model was employed to distinguish between renal recovery and non-recovery in NZB/W F1 mice (Figure 5a). The optimal cut-off point is identified as the point on the ROC curve closest to the top-left corner, representing perfect classification, as illustrated in Figure 5a. We have included this information in the statistical analysis section to enhance clarity (Page 14, lines 429-431).
- There are significant differences between the animal models in this study and human kidney disease. The authors should have clearly stated the limitations of this study.
Following your suggestion, we have included this limitation in the discussion section (Page 11, lines 305-314).
- There is no detailed experimental procedure for immunofluorescence experiments.
The detailed experimental procedures for immunofluorescence can be found in Section 4.3 of the methodology, and additional information has been included in the supplementary material (Page 13, lines 363-378).
- Editing was required to improve the English throughout the manuscript.
To enhance the English style of the manuscript, a native English speaker has reviewed it and made improvements. We have included a certificate to confirm this.

Round 2
Reviewer 1 Report
Comments and Suggestions for Authors
The authors have answered all my comments with valid reasons and incorporated all suggestions.
Reviewer 2 Report
Comments and Suggestions for Authors
The authors responded adequately to the Reviewer's comments. A revised version of the paper may be published.